# Polyaniline Nanotubes/Carbon Cloth Composite Electrode by Thermal Acid Doping for High-Performance Supercapacitors

**DOI:** 10.3390/polym11122053

**Published:** 2019-12-11

**Authors:** Jia Hui, Daoxin Wei, Jing Chen, Zhou Yang

**Affiliations:** 1Engineering Technology and Materials Research Center, China Academy of Transportation Sciences, Beijing 100029, China; huijiabuct@126.com (J.H.); weidaoxin2009@126.com (D.W.); cjtransport@163.com (J.C.); 2Department of Material Engineering, Jiangsu University of Technology, Changzhou 213001, China

**Keywords:** polyaniline nanotubes, composite electrode, thermal acid doping, flexible solid state supercapacitors

## Abstract

Carbon materials have been widely used in designing supercapacitors (SCs) but the capacitance is not ideal. Herein, we synthesize polyaniline (PANI) nanotubes on the basis of a carbon cloth (CC) through a one-step self-degradation template method, and fabricate a CC@PANI NTs-H (CC@PANI nanotubes doping at high temperature) composite electrode by thermal acid doping. The CC@PANI NTs-H electrode obviously exhibits better electrochemical performance with a gravimetric capacitance of 438 F g^−1^ and maintains 86.8% after 10,000 cycles than the CC@PANI NTs-R (CC@PANI nanotubes doping at room temperature) electrode. Furthermore, we assemble a flexible solid state supercapacitor (FSSC) device with the as-prepared CC@PANI NTs-H composite electrodes, showing good flexibility and outstanding electrochemical performances with a high gravimetric capacitance of 247 F g^−1^, a large energy density of 21.9 Wh kg^−1^, and a capacitance retention of 85.4% after 10,000 charge and discharge cycles. Our work proposes a novel and easy pathway to fabricate low-cost FSSCs for the development of energy storage devices.

## 1. Introduction

With ever-increasing demands for efficient energy, considerable efforts have been made to design and fabricate new energy conversion/storage devices [1]. Supercapacitors (SCs), as a new class of energy storage device, have received much attention over the past years due to their wide working temperature range, high power-delivery capability, fast charge-discharge rate and long cycle-stability [2]. Electrical double-layer capacitors (EDLCs) usually based on carbon materials including graphene and carbon nanotubes, but the specific capacitances are typically less than 100 F g^−1^, which largely limits their applications [2,3]. Pseudocapacitors, characterized by a series of fast and reversible redox reactions or Faradic charge transfer, exhibit a larger capacitance and energy density compared with EDLCs [4]. However, due to volumetric swelling/shrinking in the charging and discharging process, pseudocapacitive materials including conducting polymers and transition metal compounds often show poor cyclic stability [4,5].

It is a good strategy to obtain the synergetic effect of EDLC and pseudocapacitive materials in the field of designing a composite electrode [6]. Shao et al. [7] synthesized a hierarchical polypyrrole (PPy) @ layered double hydroxides (LDHs) core-shell arrays to fabricate flexible solid state supercapacitors (FSSCs), the cycling stability was improved by 15.4% after 20,000 cycles. Chen et al. [8] manufactured a composite electrode based on spherical PPy nanoparticles growing on the reduced graphene oxide (RGO)-coated carbon cloth (CC), which showed high capacitance and energy density. Yun et al. [9] assembled a wearable and all-transparent supercapacitor, consisting of an Au/Ag core-shell nanowire-embedded polydimethylsiloxane (PDMS) and WO_3_ nanotube/PEDOT: PSS thin layer, exhibiting a powerful ability of charge storage and excellent flexibility to wear.

PPy is a frequently-used conducting polymer, but there is a difference between theoretical prediction of the capacitance and practical value when it is used for supercapacitors [10]. As another important conducting polymer, polyaniline (PANI) is a highly promising electrode material for a supercapacitor due to its high pseudocapacitance, but its poor cycling stability has resulted in people making much more effort to enhance the electrochemical property in recent years [11,12,13,14]. Du et al. [15] used celery as biomass carbon precursor to prepare nitrogen-doped hierarchical porous carbon materials by combining with PANI, also displaying high specific capacitance of 402 F g^−1^ and an outstanding cycling stability. Lee et al. [16] designed a new strategy using aniline tetramers loaded on graphene oxide (AT–GO), the capacitance can reach 769 F g^−1^ at 1 A g^−1^ and still remains 581 F g^−1^ at 60 A g^−1^.

In this study, we used CC as a carbon-based substrate and grew PANI nanotubes through a one-step self-degradation template method, then activated a CC@PANI NTs composite by thermal acid doping. CC@PANI NTs-H (doping at a high temperature) composite electrodes obviously exhibit better electrochemical performances than CC and CC@PANI NTs-R (doping at room temperature) composite electrodes. Furthermore, CC@PANI NTs-H composite electrodes were used to assemble symmetric FSSCs, the gravimetric capacitance is 247 F g^−1^ and the maximum energy density reaches 21.9 Wh kg^−1^, meanwhile showing good cycling stability and favorable flexibility.

## 2. Experimental

### 2.1. Materials

Carbon cloth (CC) was purchased from Taiwan Carbon Energy technology company. Polyvinyl alcohol (PVA) was bought from Shanghai Qiangshun chemical company (Shanghai, China). Acetone, ethanol, aniline, methyl orange (MO), FeCl_3_^.^6H_2_O, (NH_4_)_2_S_2_O_8_, HNO_3_, H_2_SO_4_ and H_3_PO_4_ were purchased from Shanghai Lingfeng chemical company (Shanghai, China). Separator (NKK TF4535) was provided by Koji paper industry corporation of Japan (Tokyo, Japan).

### 2.2. Activation of CC

CC was cut into slices of the size of 1.5 × 1 cm^2^, in which the working area was 1 cm^2^ and the extra part was used as the current collector. Then CC slices were cleaned by acetone, ethanol and deionized water in an ultrasonic bath, in turn. CC was immersed into a 5 M HNO_3_ solution for 16 h, and then washed with deionized water. Finally, the activated CC was dried at 50 °C for 4 h.

### 2.3. Preparation of CC@PANI NTs

The as-prepared CC was stirred in methyl orange (MO) solution (5 mM, 20 mL) for 0.5 h, and then 2.5 mmol FeCl_3_^.^6H_2_O was added into the solution to form a FeCl_3_–MO template [17]. After 0.5 h, aniline monomer (2.5 mmol, 228 μL) was added into the system and stirred for another 24 h at room temperature. Finally, the CC@PANI NTs was obtained after washing by deionized water and ethanol.

### 2.4. Thermal Acid Doping of CC@PANI NTs

The as-synthesized CC@PANI NTs was immersed into 5 M H_2_SO_4_, and then kept in a Teflon autoclave at 100 °C for 4 h. After the reaction, the CC@PANI NTs was washed with deionized water and dried at 50 °C, named CC@PANI NTs-H. Meanwhile, we also carried out a control group to demonstrate the effect of thermal acid doping, that was doping in 5 M H_2_SO_4_ at an ambient temperature, named CC@PANI NTs-R.

### 2.5. Assembly of FSSCs

The PVA-H_3_PO_4_ gel electrolyte was prepared as follows: 3 g PVA and 30 mL deionized water were mixed and stirred at 90 °C until the gel became transparent, and cooled to room temperature [18]. Then 3 g H_3_PO_4_ was dropwise added into the gel under stirring for several hours to obtain the PVA-H_3_PO_4_ gel. The CC@PANI NTs-H electrodes were soaked into a PVA-H_3_PO_4_ gel for 10 min and then assembled together with a separator (NKK TF4535, 35 μm) to prepare FSSCs devices. The assembly devices were solidified at ambient temperature for electrochemical measurements.

### 2.6. Characterization

The electrodes were characterized by a field-emission scanning electron microscope (FE-SEM, Sigma 500, Carl Zeiss, Jena, Germany), transmission electron microscopy (TEM, JEM-2100, Jeol, Tokyo, Japan), Fourier transform infrared spectrometry analyzer (FTIR, Nicolet 6700, Thermo Scientific, Waltham, MA, USA), Raman spectroscopy (XploRA, HORIBA, Irvine, CA, USA) and x-ray photoelectron spectroscopy (XPS, PHI 5000CESCA, Perkin-Elmer, Waltham, MA, USA). Electrochemical performances such as cyclic voltammetry (CV), galvanostatic charge/discharge (GCD) and electrochemical impedance spectroscopy (EIS), were measured on an electrochemical workstation (Chenhua, CHI760E, Shanghai, China). The as-fabricated composite electrodes were tested in a three-electrode system with a 1 M H_2_SO_4_ solution as an electrolyte, Hg/Hg_2_Cl_2_ electrode and Pt wire were used as the reference and counter electrodes, respectively. The assembled FSSC devices were tested in a two-electrode system.

### 2.7. Calculations

The as-fabricated electrodes were measured in the three-electrode system. The areal capacitance (*C*_A_: mF cm^−2^) and gravimetric capacitance (*C*_g_: F g^−1^) were calculated from the GCD curves according to the following Equations:*C*_A_ = 1000(*I*Δ*t*)/(Δ*VS*)(1)
*C*_g_ = (*I*Δ*t*)/(Δ*Vm*)(2)
where *I* (A) is the discharging current, Δ*t* (s) is the discharging time, Δ*V* (V) is the voltage window, *S* (cm^2^) is the working area of electrodes, and *m* (g) is the mass loading of active materials on CC.

In addition, the FSSC devices were measured in a two-electrode system. The areal capacitance (*C*_A_: mF cm^−2^) and gravimetric capacitance (*C*_g_: F g^−1^) of the FSSCs were also calculated from GCD curves according to the following Equations:*C*_A_ = 1000(*I*Δ*t*)/(Δ*VS*)(3)
*C*_g_ = (*I*Δt)/2(Δ*Vm*_t_)(4)
where *I* (A) is the discharging current, Δ*t* (s) is the discharging time, Δ*V* (V) is the voltage window, *S* (cm^2^) is the working area of FSSCs devices, and *m*_t_ (g) is total mass loading of the single electrode in FSSCs devices.

Energy density (*E*: Wh kg^−1^) and power density (*P*: W kg^−1^) of devices were calculated according to the following Equations:*E* = (*C*_g_Δ*V*^2^)/(4 × 3.6)(5)
*P* = 3600 × *E*/Δ*t*(6)
where *C*_g_ (F g^−1^) is gravimetric capacitance of FSSCs, Δ*V* (V) is the voltage window and Δ*t* (s) is the discharging time.

## 3. Results and Discussion

### 3.1. Mechanism and Morphologies of CC@PANI NTs Composite Electrodes

The fabrication process of CC@PANI NTs composites is schematically illustrated in Figure 1. The CC substrate is first activated in HNO_3_, and then FeCl_3_ is reacted with MO on the surface of CC substrate to form nano-fibrous templates (FeCl_3_–MO). Additionally, aniline molecules are in situ polymerized under the initiating action of ammonium persulfate, and then grown on the surface of FeCl_3_–MO fibrous templates. After that, the FeCl_3_–MO fibrous templates are washed by deionized water, only PANI NTs are left. Finally, CC@PANI NTs are obtained by doping with H_2_SO_4_ at room or high temperature, named CC@PANI NTs-R or CC@PANI NTs-H.

The microstructures and morphologies of CC, PANI NTs and CC@PANI NTs-H composite materials are revealed by FE-SEM and TEM, as shown in Figure 2. Figure 2a shows the surface of CC is smooth and neat. However, a number of nanofibers are grown on the surface of CC due to the in situ polymerization of aniline, as shown in Figure 2b. To clearly observe the structure of PANI fibers in CC@PANI NTs composite, we amplified PANI NTs and characterized by FE-SEM and TEM (Figure 2c,d). Figure 2c shows that the diameter of PANI NTs is about 200–300 nm. Figure 2d demonstrates the microfiber is hollow tubular structure.

### 3.2. Characterizations of CC@PANI NTs Composite Electrodes

Figure 3a shows Fourier Transform Infrared Spectrometer (FTIR) spectrums of the CC, PANI and CC@PANI NTs-H composite. From Figure 3a, almost no characteristic peak can be found in the spectrum of CC because of few functional groups. Compared with CC, the characteristic peaks of PANI at 1578 cm^−1^ (C=C stretching vibration in Quinone), 1500 cm^−1^ (C=C stretching vibration in benzene ring), 1312 cm^−1^ (C–N stretching vibration) and 1132 cm^−1^ (N=Q=N stretching vibration), all appear but shift to low frequency range due to the effect of H_2_SO_4_ doping, in the spectrum of the CC@PANI NTs-H composite [19]. Gizdavic-Nikolaidis et al. [20] studied the FTIR spectrums of PANI doped H_2_SO_4_, an obvious shift was observed with increasing concentration of H_2_SO_4_ due to the form of depronated band.

Raman spectra in Figure 3b shows two obvious characteristic peaks at 1358 and 1604 cm^−1^, which assign disordered sp^3^ carbon (D band) and graphitic sp^2^ (G band) [21]. The intensity ratio of D and G band (*I*_D_/*I*_G_) was also calculated, 1.08 for CC, 0.98 for CC@PANI NTs-R and 0.88 for CC@PANI NTs-H, respectively. The ratio of amorphous carbon is high in CC, while the graphitic degree is enhanced after thermal doping in CC@PANI NTs-H. The existence of H_2_SO_4_ can increase ordered structure of carbon and conjugation length of PANI, which may improve the conductivity of PANI and promote transfer of ions and electrons [22].

We also analyzed the composition of the CC@PANI NTs-H by x-ray photoelectron spectroscopy (XPS), the results are displayed in Figure 3c,d. There are three sharp peaks at 532.8, 400.6 and 285.3 eV in Figure 3c, which represent O 1s, N 1s and C 1s, respectively. N element is mainly from PANI, so we analyzed N 1s spectra by fitting, as shown in Figure 3d. The split peak1 at 400.3 eV and peak 2 at 402.2 eV can be ascribed to the ground state N–H and protonated nitrogen species (N^+^) [23], respectively. The amount of intrinsic oxidation state (N^+^) of PANI equals to the value of (−N = +N^+^), which can promote protons donation and electrons transfer [22].

### 3.3. Electrochemical Performances of CC@PANI NTs Composite Electrodes

Next we measured the electrochemical performances of CC, CC@PANI NTs-R and CC@PANI NTs-H composite electrodes in 1 M H_2_SO_4_ electrolyte by the three-electrode setup. The mass loading of active materials on the electrode is about 2 mg cm^−2^. Figure 4a,c reflect CV curves of CC@PANI NTs-R and CC@PANI NTs-H electrodes at different scan rates. As shown in Figure 4a,c, the area of close curves gets larger with the increase of scan rate, while the shape of CV curves exhibit Faradic redox peaks, suggesting typical pseudocapacitive behaviors of both CC@PANI NTs-R and CC@PANI NTs-H electrodes. In addition, the GCD curves shown in Figure 4b,d also exhibit obvious pseudocapacitive behaviors, especially CC@PANI NTs-H electrode. Figure 4e,f shows comparison among CC, CC@PANI NTs-R and CC@PANI NTs-H. CC shows little area in the CV curve, correspondingly, its discharging time is also quite short, demonstrating the specific capacitance of CC is small. CC@PANI NTs-H electrode shows a larger area than CC@PANI NTs-R electrodes in Figure 4e, and the discharging time of former is also longer than latter in Figure 4f. The maximum gravimetric capacitance of CC@PANI NTs-H is 438 F g^−1^, which are larger than that of CC@PANI NTs-R (276 F g^−1^), respectively. The values are superior to most similar electrodes [18,24,25,26,27,28], as shown in Table 1. The better electrochemical performances of the CC@PANI NTs-H electrode in comparison with CC@PANI NTs-R can be attributed to acid doping at high temperature, which promotes the ordered structure of PANI and provides more sufficient protons [29].

Figure 5a shows the Nyquist plots of composite electrodes, and the inset in Figure 5b reveals the high frequency region of curves. The intercept at the real axis in the high frequency region represents the equivalent series resistance (ESR). The ESR of CC@PANI NTs-H electrode (1.64 Ω) is much smaller, compared with CC@PANI NTs-R (2.10 Ω). The results reflect a low-resistant property in CC@PANI NTs-H, suggesting its low diffusion hindrance for ions and electrons [30]. Figure 5b reveals the cycling performance of the CC@PANI NTs-H electrode, the capacitance can still maintain 86.8% after 10,000 charge and discharge cycles, which demonstrates that the poor cycling stability of PANI has been improved a lot.

### 3.4. Electrochemical Performances of Assembled FSSCs Device from CC@PANI NTs-H Flexible Electrodes

The symmetric FSSC was assembled with two CC@PANI NTs-H composite electrodes, the working area is 1 cm^2^ and the thickness of the assembled device is 0.08 cm. The electrochemical performances of the device have been studied. From Figure 6a, redox peaks are observed in CV curves of CC@PANI NTs-H FSSC, showing pseudocapacitive characteristics. Figure 6b shows the GCD curves of the device at different current densities, the asymmetrical triangle profile also suggests pseudocapacitive behaviors. The ions and electrons can easily transport in the CC@PANI NTs-H material and penetrate the gel electrolyte. As shown in Figure 6c, the capacitance retention maintains 85.4% after 10,000 cycles, indicating good cycling stability and a sturdy electrode structure. Figure 6d shows the CV curves under different bending angles at the scan rate of 100 mV s^−1^, the coincident shapes under the bending angles of 45°, 90° and 135° demonstrates good flexibility of the CC@PANI NTs-H FSSCs.

Furthermore, we calculated the areal and gravimetric specific capacitances, and listed the plots of energy density to power density. As shown in Figure 6e, the maximum areal and gravimetric specific capacitances are 494 mF cm^−2^ and 247 F g^−1^, respectively. From Figure 6f, the CC@PANI NTs-H FSSC exhibits a high energy density from 21.9 Wh kg^−1^ (at the power density of 5.0 × 10^3^ W kg^−1^) to 9 Wh kg^−1^ (at the power density of 1.2 × 10^4^ W kg^−1^). The values are comparable with other congeneric FSSCs devices [31,32,33,34,35], as can be seen in Table 2.

## 4. Conclusions

In conclusion, we fabricated a novel CC@PANI NTs-H flexible electrode through a one-step self-degradation template method and thermal acid doping process, the gravimetric capacitance can reach 438 F g^−1^ and maintain 86.8% after 10,000 cycles, the unique thermal acid doping method obviously exhibits better electrochemical performances than doping at room temperature. Moreover, we assembled a FSSC device with the as-prepared CC@PANI NTs-H composite electrodes, delivering favorable flexibility and excellent electrochemical performances with a high gravimetric capacitance of 247 F g^−1^, a large energy density of 21.9 Wh kg^−1^, and a capacitance retention of 85.4% after 10,000 cycles. This study provides a new direction for fabricating a low-cost and high-performance FSSCs in the energy storage field.

## Figures and Tables

**Figure 1 polymers-11-02053-f001:**
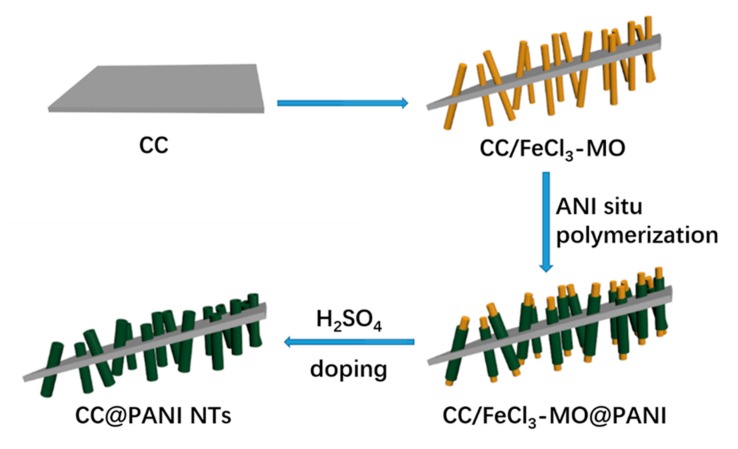
Schematic illustration of the fabrication process of the CC@PANI NTs.

**Figure 2 polymers-11-02053-f002:**
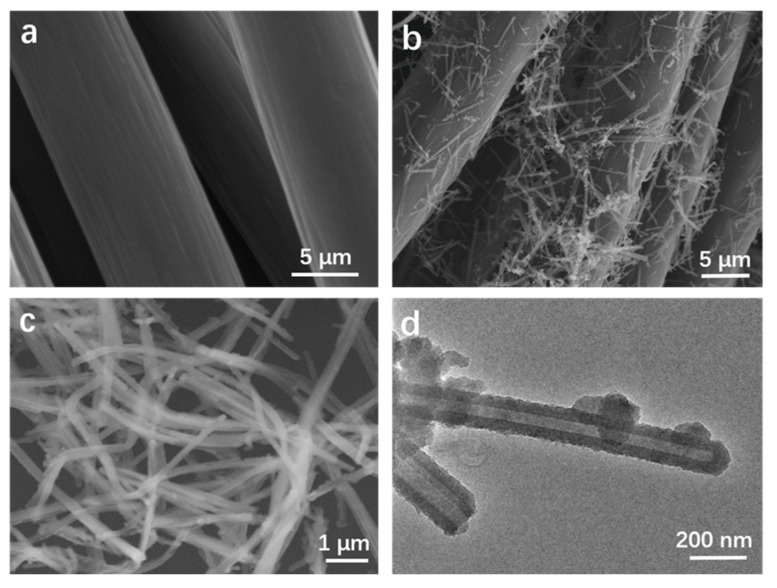
FE-SEM images of (**a**) CC, (**b**) CC@PANI NTs-H, (**c**) PANI NTs from CC@PANI NTs-H. (**d**) TEM images of PANI NTs from CC@PANI NTs-H composite.

**Figure 3 polymers-11-02053-f003:**
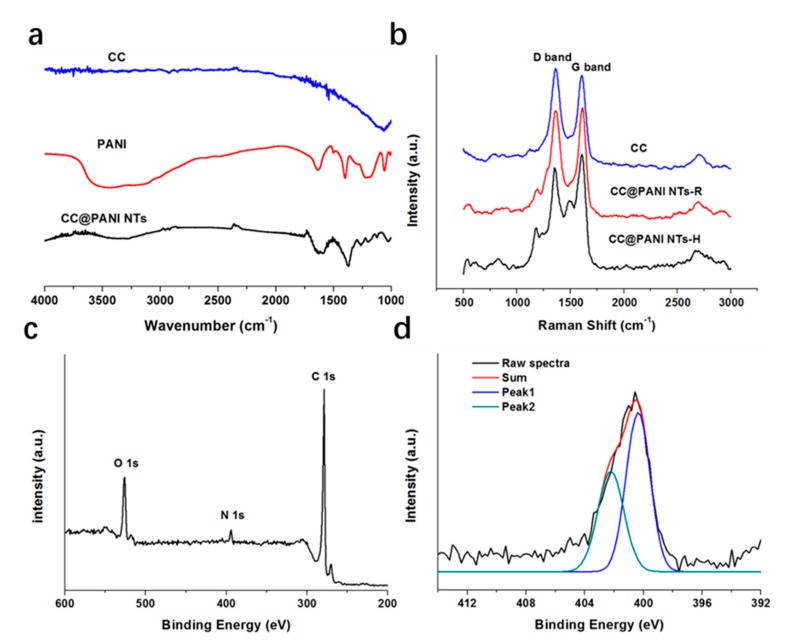
(**a**) FTIR spectra of the CC, PANI and CC@PANI NTs, (**b**) Raman spectra of the CC, CC@PANI NT-R and CC@PANI NT-H, (**c**) XPS survey spectra (C 1s, N 1s and O 1s) of CC@PANI NT-H, (**d**) fitting of N 1s spectra.

**Figure 4 polymers-11-02053-f004:**
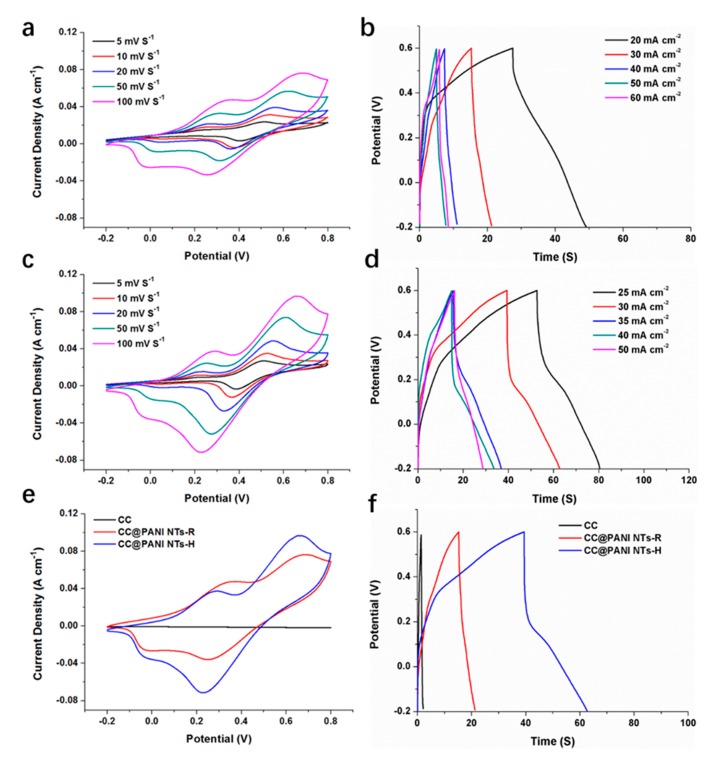
CV curves (**a**) and GCD curves (**b**) of CC@PANI NTs-R electrodes. CV curves (**c**) and GCD curves (**d**) of CC@PANI NTs-H electrodes. CV curves at 100 mV s^−1^ (**e**) and GCD curves at 30 mA cm^−2^ (**f**) of CC, CC@PANI NTs-R and CC@PANI NTs-H electrodes.

**Figure 5 polymers-11-02053-f005:**
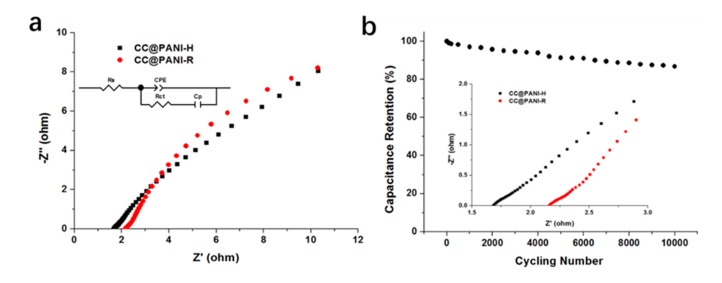
(**a**) Nyquist plots of the CC@PANI NTs-R and CC@PANI NTs-H composite electrodes, and the inset in top left corner is equivalent circuit. (**b**) Cycling performance of CC@PANI NTs-H electrode at 30 mA cm^−2^, the inset in bottom is the magnifying part in the high frequency region of Nyquist plots.

**Figure 6 polymers-11-02053-f006:**
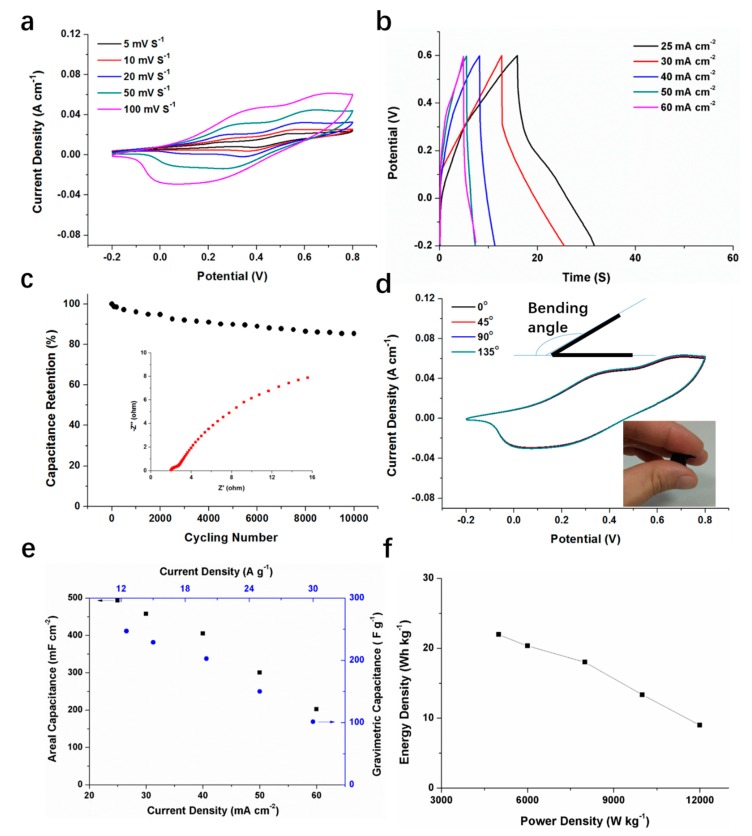
The electrochemical performances of the symmetric CC@PANI NTs-H FSSCs devices. (**a**) CV curves measured at various scan rates from 5 to 100 mV s^−1^. (**b**) GCD curves measured at different current densities from 25 to 60 mA cm^−2^. (**c**) Capacitance retention after 10,000 cycles at 60 mA cm^−2^, the inset is Nyquist plots. (**d**) CV curves under different bending angles at the scan rate of 100 mV s^−1^, the inset in right bottom is the supercapacitor under the bending effect. (**e**) Areal (black squares) and gravimetric specific capacitances (blue dots) at different current densities. (**f**) Energy densities at different power densities.

**Table 1 polymers-11-02053-t001:** Comparison of gravimetric capacitance with other PANI-related electrodes.

Electrodes	Maximum Gravimetric Capacitance (F g^−1^)
PANI-ZIF-67-CC [18]	371
RGO/V_2_O_5_/PANI [24]	273
PANI/Silica Self-Aggregates [25]	218.75
PANI@Nanodiamond-Graphene [26]	150.20
Carbon Fiber/PANI [27]	350
Multiwalled Carbon Nanotube/PANI [28]	333
CC@PANI NTs-H (This work)	438

**Table 2 polymers-11-02053-t002:** Comparison of energy density with other PANI-based SC devices.

FSSCs Device	Maximum Energy Density (Wh kg^−1^)
RGO/Cu_2_O/PANI [31]	18.95
RGO/PANI [32]	17.6
PANI/Modified OGH [33]	11.3
PANI/Mn_3_(PO4)_2_ [34]	14.7
FD-Fe-PANI@s-MCNTs [35]	13.55
CC@PANI NTs-H (This work)	21.9

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
