# Peer review of "Polyaniline Nanotubes/Carbon Cloth Composite Electrode by Thermal Acid Doping for High-Performance Supercapacitors"

_polymers, 2019, doi:10.3390/polym11122053_

Round 1

Author Response

Dear Editor and Referee,

Thank you very much for your comments on our manuscript (ID: Polymers-661642, entitled as “Polyaniline nanotubes /Carbon cloth composite electrode by thermal acid doping for high-performance supercapacitors”). Here below are the point-by-point responses to the comments from the referee, and highlighted in red in the revised manuscript.

Question 1: Experimental procedure:

please provide the method to calculate the specific capacitance of the electrode. Please also provide the typical mass of the active materials on the electrode. Are all the electrodes have similar mass during the charge and discharge test? please provide the method to calculate the specific capacitance and energy density of the solid-state supercapacitor device. Please also provide the typical mass of the active materials on the solid-state device.

Answer 1: Thanks for your comment. We have added the method to calculate the specific capacitance and energy density in the revised manuscript. The typical mass of the active materials on the electrode is about 2 mg cm-2, and the mass of the active materials on the single electrode in the solid-state device is also about 2 mg cm-2, all the electrodes have similar mass during the charge and discharge test.

Question 2: Authors only cite the work that show poorer performance than what they have obtained. Some works on polyaniline based electrode have reported higher capacitance such as Adv. Energy Mater. 2014, 1400781 and Electrochim. Acta 65 (2012) 190.

Answer 2: Thanks for your suggestion. We have cited the papers recommended by the referee in the revised manuscript.

Question 3: The fitting of N1s XPS spectra is not properly fitted. Authors might want to check this reference on how it could be done J. Mater. Chem., 2012, 22, 23921. The role of various state of Nitrogen can be discussed.

Answer 3: Thanks for your comment. We have checked and read the reference you recommended, and fitted N1s XPS spectra again. We discussed the role of various state of Nitrogen, and some sentences have been revised as follows: “The split peak1 at 400.3 eV and peak2 at 402.2 eV can be ascribed to the ground state N-H and protonated nitrogen species (N+), respectively. The amount of intrinsic oxidation state (N+) of PANI equals to the value of (-N=+N+), which can promote protons donation and electrons transfer.”

Question 4: I personally think the language of this manuscript need to be improved. The English in results and discussion is all in the past tense form. Not every sentences in the results and discussion need to be in the past tense, this is not required when discussing the results.

Answer 4: We repeatedly checked and revised our manuscript, and some sentences in the results and discussion were revised to be the present tense in the revised manuscript.

Question 5: Some sentences in the introduction can be improved for a better clarity

Page 1 line 41 “the cycling stability was improved a lot”. Authors can provide the exact number of the stability improvement Page 2 line 47 “Ppy is a usual conducting polymer”. What does the word “usual” mean here? Page 2 line 50 “the drawback of poor cycling stability delivers people making much efforts to enhance the electrochemical property of PANI”. I am not sure about the meaning of this sentence. Page 3 line 107 “Then aniline molecules were SITU polymerized. The word should be in situ polymerized.

Answer 5: a. Thanks for your suggestion, we provided the exact number of the stability improvement according to the cited reference in the revised manuscript.

“Usual” means frequently-used, we revised it to “frequently-used” in the revised manuscript. We are so sorry for that, now we have already revised the original sentence to “its poor cycling stability delivers people making much efforts to enhance the electrochemical property”. Thank you, now we revised the sentence to be “Then aniline molecules were in situ polymerized” in the revised manuscript.

Hopefully our responses and explanations above to referee’s comments can make the manuscript understandable and readable clearly.

Thank editor and the referee very much for the kind advices. Please feel free to contact us if you have any further questions.

Yours Sincerely,

Dr. Zhou Yang

School of Material Engineering,

Jiangsu University of Technology,

Changzhou 213001, P. R. China.

Tel: +86-519-86953292.

Reviewer 2 Report

The manuscript “Polyaniline nanotubes /Carbon cloth composite electrode by thermal acid doping for high-performance supercapacitors” reported polyaniline (PANI) nanotubes on the basis of carbon cloth (CC) synthesized via a one-step self-degradation template method, and followed by thermal acid doping. The flexible solid state supercapacitor (FSSC) performance of the as-prepared composite electrode was also investigated. The authors have provided solid data to back up the conclusions in most cases. However, when reading the manuscript some questions arise, therefore some complementary information and revision should be taken into account before being published in “Polymers”. 1. The abbreviations should be defined before using, such as “NTs-H” and “NTs-R” in Abstract. 2. The synthesized PANI amount on one piece of CC electrode should be measured and provided. 3. In Figure 1 and its related discussion, the phrase “situ polymerization” is not clear. It is better to clarify it is either in-situ or out-situ. 4. In section 3.2, the authors claimed that the characteristic peaks of PANI shifted to low frequency range due to the effect of H2SO4 doping. However, why and how this effect happened should also be discussed. 5. The GCD curves in Figure 4 and 6 show much shorter discharging time that corresponding charging time, suggesting low coulombic efficiencies. From the CV profiles, the discharging cut-off voltage of -0.2 V is suggested for the GCD tests. 6. The inset of Figure 5a is not clear. 7. The major problem of this manuscript is that most of the texts are focusing on descriptions of the figures but without deeply discussion and explanation. 8. The English writing in manuscript needs to be checked very carefully before submission since the meaning of many sentences cannot be understood by the improper use of English or ambiguous description.

Author Response

Dear Editor and Referee,

Thank you very much for your comments on our manuscript (ID: Polymers-661642, entitled as “Polyaniline nanotubes /Carbon cloth composite electrode by thermal acid doping for high-performance supercapacitors”). Here below are the point-by-point responses to the comments from the referee, and highlighted in red in the revised manuscript.

Question 1: The abbreviations should be defined before using, such as “NTs-H” and “NTs-R” in Abstract.

Answer 1: Thanks for your comments. We added the full name in Abstract in the revised manuscript.

Question 2: The synthesized PANI amount on one piece of CC electrode should be measured and provided.

Answer 2: Thanks for your comments. We gave the total mass loading of PANI on CC in the revised manuscript.

Question 3: In Figure 1 and its related discussion, the phrase “situ polymerization” is not clear. It is better to clarify it is either in-situ or out-situ.

Answer 3: Thanks for your comments. We revised the phrase “situ polymerization” to be “in situ polymerized” in the revised manuscript.

Question 4: In section 3.2, the authors claimed that the characteristic peaks of PANI shifted to low frequency range due to the effect of H2SO4 doping. However, why and how this effect happened should also be discussed.

Answer 4: We discussed the effect of H2SO4 doping for FTIR spectrums in the revised manuscript as follows: “Gizdavic-Nikolaidis et al studied the FTIR spectrums of PANI doped H2SO4, an obvious shift was observed with increasing concentration of H2SO4 due to the form of depronated band.”

Question 5: The GCD curves in Figure 4 and 6 show much shorter discharging time that corresponding charging time, suggesting low coulombic efficiencies. From the CV profiles, the discharging cut-off voltage of -0.2 V is suggested for the GCD tests.

Answer 5: Thanks for your suggestions. We did electrochemical measurements with cut-off voltage is -0.2 V for the GCD tests again according to your suggestion, and the coulombic efficiencies were enhanced as expected, the results are also revised in the revised manuscript.

Question 6: The inset of Figure 5a is not clear.

Answer 6: The inset of Figure 5a maybe too small to watch, we moved it to Figure 5b and magnified it, now it is clearer in the revised manuscript.

Question 7: The major problem of this manuscript is that most of the texts are focusing on descriptions of the figures but without deeply discussion and explanation.

Answer 7: Thanks for your comments. We discussed more deeply and gave explanations around the results in the revised manuscript.

Question 8: The English writing in manuscript needs to be checked very carefully before submission since the meaning of many sentences cannot be understood by the improper use of English or ambiguous description.

Answer 8: Thanks for your comments. We carefully checked our English and improved the writing in the revise manuscript.

Hopefully our responses and explanations above to referee’s comments can make the manuscript understandable and readable clearly.

Thank editor and the referee very much for the kind advices. Please feel free to contact us if you have any further questions.

Yours Sincerely,

Dr. Zhou Yang

School of Material Engineering,

Jiangsu University of Technology,

Changzhou 213001, P. R. China.

Tel: +86-519-86953292.

Round 2

Reviewer 2 Report

All the questions and comments raised by the reviewers were legitimately explained and revised, except for the phrase “situ polymerization” in Fig. 1. The accuracy and detail of the manuscript were also improved further after revision. Therefore, the revised manuscript will be suitable publication in "Polymers" after a minor revision.